# Effect of Tempering Conditions on Secondary Hardening of Carbides and Retained Austenite in Spray-Formed M42 High-Speed Steel

**DOI:** 10.3390/ma12223714

**Published:** 2019-11-11

**Authors:** Bowen Liu, Tian Qin, Wei Xu, Chengchang Jia, Qiuchi Wu, Mingying Chen, Zhe Liu

**Affiliations:** 1Institute for Advanced Materials and Technology, University of Science and Technology Beijing, Beijing 100083, China; xuweicool@126.com; 2School of Materials Science and Engineering, University of Science and Technology Beijing, Beijing 100083, China; TianQin_ustb@163.com (T.Q.); wqc961007@163.com (Q.W.); 15266247809@163.com (M.C.); sktcllz123@126.com (Z.L.)

**Keywords:** spray formed M42 HSS, tempering, secondary hardening carbides, EBSD

## Abstract

In this study, the effect of tempering conditions on microstructure, grain size, and carbide phase compositions of spray-formed high-speed steel after quenching at 1180 °C was studied. The influence of carbide phase, size of carbides, and retained austenite content on secondary hardening of the steel was analyzed by field emission scanning electron microscope (FESEM), transmission electron microscope (TEM), electron backscattered diffraction (EBSD), and differential scanning calorimetry (DSC); the hardness, microhardness of carbide, and bending strength were tested. The results show that M_3_C, M_6_C, M_7_C_3_, and MC carbides may precipitate at different tempering temperatures and the transformation of the retained austenite can be controlled by tempering. The phase composition of carbides, microstructure, and retained austenite content strongly influences the performance characteristics of M42 high-speed steel after tempering. In contrast, the secondary carbides produced by tempering thrice at 540 °C are mainly M_6_C carbides rich in W and Mo elements, and the content of retained austenite is effectively reduced. At this stage, the Rockwell hardness reaches 67.2 HRC, bending strength reaches 3115 MPa, and the properties and microstructure are optimal.

## 1. Introduction

High-speed steel (HSS) is a type of Fe-C-X high alloy and high carbon content steel, where X represents Mo, W, Cr, and another alloy [1,2]. It has been used as a tool material for almost 100 years, and is still used to manufacture taps, dies, cutters, drills, and other cutting equipment [3,4,5]. Its excellent performance in the cutting process is mainly due to a combination of strengthening of the martensite matrix and secondary hardening of carbides [6]. In terms of chemical composition, HSS usually contains a high content of W, Mo, Cr, V and other strong carbide-forming elements. The dispersion of alloy carbides is the basis of HSS’ hot hardness, wear resistance, and bending strength. Cast HSS has problems in alloy element segregation and coarseness of carbides, among others. Hot deformation is usually required to break the carbide. However, the carbide band produced by hot deformation damages the anisotropy of the material, thus impairing performance [7,8,9]. This is where powder metallurgy (PM) comes into play. It is a three-step (powder making, forming, and sintering) preparation process, which can be realized in several ways. Rapid cooling in the production process can result in a fine structure and reduce segregation, thus improving the mechanical properties of the materials [10,11,12]. The application of this technology to high-speed steel can produce a material with fine structure and uniform carbide distribution. Therefore, compared to traditionally cast HSS, PM HSS has better mechanical properties. However, its complex process and low production efficiency of hot isostatic pressing (HIP) greatly increases production costs [13,14]. In recent decades, spray forming, due to its high cooling rate and unique one-step forming process, has gradually enabled the low-cost production of HSS with low-level alloy element segregation and carbide dispersion [15,16,17,18,19].

AISI M42 HSS (EU designation: HS 2-9-1-8) has high bending strength, hot hardness, and good toughness due to the high alloy content of Mo, W, V, Cr, and Co (nearly 30%). However, it also causes the transformation of complex carbides. Heat treatment can, thus, effectively optimize the matrix structure and carbide phase composition (carbide type, size, distribution, etc.) of HSS, playing a decisive role in the optimization of its mechanical properties [20,21]. At the same time, the high alloy content holds on to the retained austenite, seriously affecting the mechanical properties of the product. Most of the retained austenite can be transformed into martensite by double or triple tempering [22], most often under 470 °C [23]. The structure and carbide transformation of traditional cast HSS with heat treatment have been extensively analyzed by researchers. Mingjia Wang et al. studied the decomposition of martensite, precipitation characteristics of carbonitride, and stability of the retained austenite in HSS during tempering [24]. Xuefeng Zhou et al. put forward a new approach for refining of carbides dimensions in M42 super-hard HSS by increasing the cooling rate and spheroidizing treatment, which can ultimately improve the supersaturation of alloy elements in martensite [25]. The types of carbides [26], change of carbide precipitation and structure after cryogenic treatment [27,28], and precipitation model of carbides during heat treatment have been studied [29,30,31]. Scholars have also analyzed the heat treatment behavior of PM S390 steel [32]. M. Godec et al. observed the carbide transformation in M42 annealing process by in-situ EBSD [3]. Their research provides a new way to analyze grain growth and phase transformation processes. However, little research has been conducted on the transformation of microstructures and carbides during the heat treatment of spray-formed HSS. 

This paper focuses on the effect of tempering temperature on microstructure, carbide transformation, size control, and retained austenite content in M42 spray-formed HSS. The phenomenon of secondary hardening of carbides by precipitation (tempering) and strengthening mechanism of the spray-formed M42 HSS were revealed by using XRD, FESEM, EBSD, and DSC.

## 2. Experiment

### 2.1. Material and Specimen Preparation

The composition of spray-formed M42 HSS, supplied by Advanced Technology & Materials Co., Beijing, China, is shown in Table 1. The HSS samples studied in this paper were processed by spray forming and subsequent precision forging. A tube resistance furnace was used to study structure transformation during the heat treatment process; thus, the size of the heat treatment sample was 5 × 10 × 20 mm. After quenching, one of the samples was prepared for DSC analysis—it was cut using a wire (sample size Ø × mm) and polished by a diamond. The other samples were further tempered, and their microstructure and properties were analyzed. In order to observe transformation of the carbide structure and retained austenite after tempering at different temperatures, the samples were prepared by diamond polishing and then electropolishing (20% H_2_SO_4_ + 80% Methanol).

### 2.2. Tempering

The sample was placed in alumina powder in an argon atmosphere to ensure uniform heating and avoid oxidation and decarburization. It was then heated to 850 °C at a heating rate of 10 °C/min for 30 min, and then heated to 1180 °C for 5 min. This was followed by oil quenching and subsequent cooling to room temperature. The quenching process was outlined by the authors in an earlier work [33]. After complete cooling, the sample was reheated to specific tempering temperature Tt in an argon atmosphere, and cooled after holding for 60 min; this was repeated thrice until the sample was at room temperature. In order to study the secondary hardening mechanism, tempering temperature was set as 200–700 °C (one point for every 100 °C), and further set to 520 °C, 540 °C, and 560 °C to explore the optimal hardness value. It is worth noting that the transformation of carbides at high temperatures is dependent on the change in oxygen content; thus, argon atmosphere (flow rate: 50 mL/min) was adopted in the experiment to ensure accuracy of the experimental data.

### 2.3. Analysis

The phase and retained austenite of M42 high-speed steel were analyzed by Dmax-RB type 12 KW rotary anode X-ray diffraction (XRD) machine, fitted with CuKα radiation (wavelength was 0.15406 nm) at a scanning rate of 1°/min. The microstructure of the steel after heat treatment was observed by using a cold field Emission Scanning Electron Microscope (SU8010, Hitachi Co., Japan) with Link-860 Energy Spectrometer. The Rockwell hardness HRC of the sample was determined by a HSRU-45 Hardness Tester. The bending strength test was carried out on an INSTRON universal material test machine at room temperature, with sample size of 3 mm × 3 mm × 20 mm, and deformation rate of 0.002 s^−1^. The microhardness of the carbide was carried out on Wolpert Wilson Instruments 401MVD-VD683 and the zone, including carbide of M42, was tested. Five points on each sample were tested for hardness, bending strength, and microhardness testing. Carbide size and volume fraction were calculated by Image Pro Plus software. The calculation formula of the retained austenite was as follows, derived according to the Chinese National Standard GB 8362-87:(1)$$V_{A}=\frac{1-V_{C}}{1+G\times I_{M\left(\mathrm{hkl}\right)}/I_{A\left(\mathrm{hkl}\right)}}$$
where, V_A_ is austenite volume fraction, V_c_ is carbide volume fraction, G is a related parameter, I_M(hkl)_ and I_A(hkl)_ are accumulated diffraction intensity of the martensite and austenite crystal surface, respectively.

FEG SEM JEOL 6500F field emission scanning electron microscope with energy-dispersive spectroscopy (INCA X-SIGHT LH2-type detector, INCA ENERGY 450 software) and EBSD (HKL Nordlys II EBSD camera, using Channel 5 software) were used for imaging and EBSD analysis. Working parameters of the instrument were: current 15 kV and 1.5 nA, tilt angle of 72°, and scanning step length of 0.2 μm.

## 3. Results

### 3.1. Microstructure of M42 Evolution

The microstructure of M42 HSS is shown in Figure 1 under different tempering conditions. It can be seen that the carbides are evenly distributed in the matrix. Compared to the traditional cast HSS, the carbide of spray-formed HSS is smaller and of a more regular shape; no fishbone-like carbides were visible. The results show that the sample had a martensite matrix after tempering under 600 °C, and the grain size of the martensite became smaller and the microstructure finer with the increase of tempering temperature (Figure 1a–e). Retained austenite can be seen at the martensite grain boundaries (shown as Figure 1b,d). When the tempering temperature was higher than 600 °C, the matrix structure transforms into homogeneous ferrite (Figure 1g,h). At the same time, the intra-granular nano-sized MC carbides precipitated.

### 3.2. Evolution of Mechanical Properties

The Rockwell hardness and bending strength of M42 high-speed steel, under 1180 °C oil quenching and different tempering temperatures, are shown in Figure 2. A secondary hardening phenomenon is visible—Rockwell hardness reaches a maximum value of 67.20 HRC and bending strength reaches 3115 MPa at a tempering temperature of 540 °C. Secondary hardening of M42 high-speed steel mainly occurs between 500 and 550 °C, and the peak appears at around 540 °C. The change in mechanical properties of M42 can be divided into three stages: (1) 200–300 °C—with increase of tempering temperature, the hardness and bending strength increases rapidly; (2) 300–540 °C—with slow optimization of mechanical properties, the Rockwell hardness increases from 66.25 HRC to 67.20 HRC; (3) 540–700 °C—with the increase of tempering temperature, the mechanical properties gradually deteriorate and the hardness slowly decreases to 62.48 HRC at 540–600 °C, then rapidly decreases to 43.30 HRC above 600 °C.

### 3.3. Evolution of Carbides

The phase composition of carbides is complex with the addition of transition metals such as W, Mo, V, Co, Cr, and Mn. It is possible to generate MC, M_6_C, M_2_C, M_3_C, M_23_C_6_, M_7_C_3_, and other carbides. Figure 3 shows the XRD patterns of M42 high-speed steel after tempering thrice at different temperatures. It can be seen that the matrix is mainly composed of tempered martensite. There is a certain amount of retained austenite in the matrix after heat treatment because of the high content of alloy elements in M42 HSS. Under different tempering conditions, the main carbide was found to be M_6_C. Due to its incomplete dissolution in the quenching process, M_6_C carbides were retained in different tempering processes. However, peak values of M_3_C carbides appeared at low tempering temperatures, and disappeared as the temperature rose above 400 °C.

In order to further analyze the effect of tempering temperature on carbide transformation, the alloy composition, size, and microhardness of carbides at different tempering temperatures were statistically analyzed (shown in Figure 4). When the tempering temperature was below 400 °C, due to the low diffusion rate, the amount of carbide that precipitated from the matrix was not much; thus, the statistics of carbide size and composition were mostly focused on incomplete dissolution in the quenching process. As shown in Figure 4a, there was little change when the tempering temperature was below 400 °C. With the increase of tempering temperature, the composition of alloy elements in the carbide saw two changes: (1) The content of V and Cr decreased. As is shown in Figure 1f–h, there were some fine intra-granular carbide precipitates, analyzed as VC or CrC. With MC intra-granular carbide precipitates, the Cr and V elements in the carbide decreased correspondingly. (2) The content of Mo and W increased. When the tempering temperature rose above 500 °C, a large amount of M_6_C type carbide was precipitated. With the rise in temperature and diffusion rate, the precipitation of carbides rich in Mo and W became easier.

As shown in Figure 4b, the changing trend of carbide microhardness is similar to that of the Rockwell hardness of M42 HSS, as shown in Figure 2. This reveals that the dispersion strengthening of carbides plays an important role in the secondary hardening of M42 HSS. The changes in microhardness of carbides leads to the hardness change in M42. The changes in microhardness of carbides come from two aspects: (1) composition change; (2) carbide phase composition change. On the one hand, with the increase in tempering temperature, a large number of W, Mo, and other elements are dissolved in the quenching process precipitate to form a carbide with high hardness. On the other hand, when the tempering temperature is low, M_3_C carbide exists, and its strength is inferior to that of M_6_C carbide. As the temperature continues to rise above 600 °C, it is speculated that there may be other types of carbide transformations, and the hardness of the carbide decreases obviously. The size change shows that when tempered at 200–400 °C, the diffusion ability of elements is relatively low, so most of them precipitate along the undissolved M_6_C type carbide. With the increase in temperature, the diffusion rate increases, and carbide size increases due to higher nuclear number densities and nucleation rates. When tempering at 400–600 °C, with the increase in tempering temperature, more carbides are dispersed and precipitated in the matrix. Therefore, the size of the carbides decreases with increase in temperature. The standard deviation also gradually decreases, reflecting the decrease of large carbides. However, when the temperature is higher than 600 °C, the carbide gradually coarsens, leading to size increase with temperature. Standard deviation, thus, decreases with increase in tempering temperature. The results show that large carbides decrease, gradually forming a small and dispersed state.

In order to understand why carbide microhardness decreases with tempering temperatures above 600 °C, considering that XRD cannot show all the phase changes, EBSD phase analysis for samples tempered at 500–700 °C was carried out. On comparison with the standard cards of W_3_Fe_3_C (227, Fd3-m) and Cr_7_C_3_ (62, Pnma), it was found that carbide transformation takes place with increase in tempering temperature. On tempering below 600 °C, the main carbide is M_6_C; its surface is smooth, as is seen in the band contract diagram. When tempering at 600 °C, as shown in Figure 5c, the transformation starts from the boundary of the carbide and martensite, and then develops to the interior. The original carbides are all uniform particles, transformed into multiple particles—some are still M_6_C, while the others are M_7_C_3_ carbides. In contrast, another carbide particle in Figure 5c is seen as being a complete M_7_C_3_ carbide—a complete particle with a smooth surface. As the tempering temperature increases to 700 °C, the carbides become M_6_C, and M_7_C_3_ carbides coexist in the matrix. The M_7_C_3_ carbide is a metastable phase, which has a destructive effect on the mechanical properties of M42 HSS after tempering. Therefore, as shown in Figure 2, the hardness and bending strength decreases. At the same time, referring to Figure 4b, it was also noted that the microhardness of carbides begins to decrease significantly when the tempering temperature rises above 600 °C.

In order to further verify the carbide transformation process, the M42 HSS sample quenched at 1180 °C was analyzed by DSC, as shown in Figure 6. After quenching, a heating rate of 1 °C/min was adopted to explore the carbide transformation process in the tempering process, and there is still a certain gap in the real tempering process. Figure 6 shows small endothermic peaks at 177.61 °C, 316.07 °C, and 517.78 °C, respectively, corresponding to the precipitation of M_3_C, M_6_C, and M_7_C_3_ carbides. This process is in accordance with the XRD patterns of M42 (Figure 3) and the analysis results of EBSD (Figure 5). With the increase in tempering temperature, M_3_C carbides precipitate in the matrix tempered below 400 °C, and M_6_C carbides precipitate at 400 °C and above. With the increase in temperature, the amount of precipitation gradually increases. When the tempering temperature rises above 550 °C, M_7_C_3_ carbides begin to precipitate, especially above 600 °C.

### 3.4. Effect of Tempering Conditions on Retained Austenite

As a high-alloy HSS, in the general heat treatment process, there is a lot of retained austenite in M42 after quenching. Tempering is necessary to reduce the content of retained austenite. The content of retained austenite at different tempering temperatures is calculated, due to the results of XRD, as shown in Figure 7. It can be seen that the content of retained austenite decreased from 27.99% at 200 °C to 10.04% at 700 °C. With the increase in tempering temperature, the content of retained austenite decreased significantly, and decreased to a low point at 550 °C. The tempering temperature continued to increase with little change in the content of retained austenite. Proper tempering temperature can effectively reduce the content of retained austenite, but not eliminate it.

In order to further analyze the transformation and microstructure of the retained austenite, the TEM morphology and selected electron diffraction results of M42 HSS after tempering at different temperatures are shown in Figure 8a. The matrix has retained austenite after tempering at 550 °C, distributed at the edge of the martensite grain. During tempering, some of the retained austenite transforms into martensite, and some remains at the grain boundary. As the tempering temperature increases to 700 °C, it can be seen from Figure 8b that the matrix of M42 has changed into ferrite (as seen in Figure 1). Most carbides in M42 HSS are larger than 1 μm, which cannot be observed in TEM. However, nucleation of the M_6_C type carbide is still observed at the grain boundary of the matrix structure, which is not fully grown. It can be seen that the M_6_C type carbide is mainly nucleated at the martensite/ferrite grain boundary.

## 4. Discussion

The secondary hardening of spray formed M42 HSS occurs after tempering. It can be seen from the above that the causes of tempering temperature affecting the mechanical properties of M42 HSS are mainly divided into three aspects: (1) matrix structure; (2) content of retained austenite; (3) influence of carbide phase composition and size.

After quenching at 1180 °C and tempering, the matrix structure changes with increase in tempering temperature. When tempered below 600 °C, with increase in tempering temperature, the dissolved carbon in the martensite gradually precipitates and the martensite structure is obviously refined. The refined martensite promotes the fine grain strengthening of the M42 HSS matrix, and improves the hardness and bending strength. When the tempering temperature rises above 600 °C, the matrix structure changes into equiaxed ferrite + retained austenite. Compared with tempered martensite, the strength of ferrite matrix decreases obviously, making the mechanical properties of M42 decrease rapidly above 600 °C.

As a high-alloy steel, retained austenite exists in M42 HSS at different temperatures. Compared with martensitic, retained austenite has a destructive effect on the mechanical properties of M42. With the increase in tempering temperature, the content of retained austenite gradually decreases. When the temperature is higher than 550 °C, the content of retained austenite gradually decreases to the lowest value. Therefore, the mechanical properties of M42 HSS gradually improve with increase in tempering temperature, until it rises over 550 °C, after which they gradually decrease.

Due to the high carbon content of M42, the carbide cannot be completely dissolved during quenching, so there is a certain amount of M_6_C carbide retained at different tempering temperatures. Considering the high alloy content of M42 HSS, carbide precipitation is complex. M_6_C carbide is the main dispersion phase in secondary hardening, with optimal mechanical properties in all aspects, which can play an important role in dispersion strengthening. According to the results of XRD, EBSD, microstructure, and DSC, with increase in tempering temperature, the phase composition of carbide precipitate becomes M_3_C, M_6_C, MC and M_7_C_3_. When the tempering temperature is low, M_3_C carbide is precipitated from M42 HSS with low microhardness. When tempered above 400 °C, a large amount of M_6_C carbides are precipitated. The microhardness of the carbide is significantly improved, as is the Rockwell hardness and bending strength of M42 HSS. When tempered above 500 °C, nanosized intra-granular MC carbide rich in V and Cr begin to precipitate, further strengthening the matrix. When tempered above 600 °C, the M_7_C_3_ carbides begin to gradually precipitate in the matrix, from the edge of the M_6_C carbides to the interior. There also simultaneously exist carbide particles composed of M_7_C_3_ carbides and M_6_C carbides. Until 700 °C, the coexistence of large M_7_C_3_ carbides and M_6_C carbides is basically complete. As metastable phase, the microhardness of M_7_C_3_ carbides is far lower than that of M_6_C carbides. From Figure 4b, it can be seen that with decrease in the microhardness of the carbide, the hardness and bending strength of the matrix are obviously reduced. Carbide size changes slightly and increases when tempering below 400 °C. Because of the low tempering temperature, the diffusion rate is low, and the carbide precipitates along the undissolved M_6_C. With increase in tempering temperature, the precipitation is accelerated, the particles become uniform and dispersed, and the size is gradually reduced, which corresponds to secondary hardening. When tempering is done above 600 °C, the carbides tend to gradually coarsen and the overall strength of M42 is further reduced. Statistics on standard deviation of size show that with increase in tempering temperature, large carbides are gradually reduced and size distribution becomes gradually uniform, showing dispersion distribution. On the other hand, with increase in tempering temperature, W and Mo content gradually increases, and the carbides harden. At the same time, the content of V and Cr in bulk carbides decreases correspondingly, due to the precipitation of MC carbides. Secondary hardening of the tempering process is closely related to carbide precipitation, size, and distribution. When the tempering temperature is about 540 °C, fine M_6_C carbides are found dispersed in M42 HSS, and nanosized intragranular MC carbides begin to precipitate, corresponding to the secondary hardening phenomenon.

## 5. Conclusions

(1)The tempering process has an impact on secondary hardening on spray-formed M42 HSS by refining martensite, reducing the retained austenite content, and controlling carbide phase composition and size. The best tempering process is three times at 540 °C for one hour each time. The Rockwell hardness of M42 HSS reaches 67.2 HRC and the bending strength reaches 3115 MPa, which is higher than the required values for cooperative enterprises (Rockwell hardness more than 65 HRC and bending strength more than 3000 MPa).(2)With the increase in tempering temperature, the martensite is refined, and the retained austenite is gradually transformed into martensite. The content of retained austenite is reduced from 27.99% at 200 °C to 10.04% at 700 °C, which increases the strength of M42 HSS. When tempering above 600 °C, the martensite matrix is gradually transformed into ferrite, and the strength of the M42 decreases.(3)Secondary hardening during tempering is closely related to the precipitation and transformation of carbide. With the increase in tempering temperature, the phase composition of carbides undergoes precipitation of M_3_C-M_6_C-M_7_C_3_, and the corresponding microhardness of the carbides first increases and then decreases.

## Figures and Tables

**Figure 1 materials-12-03714-f001:**
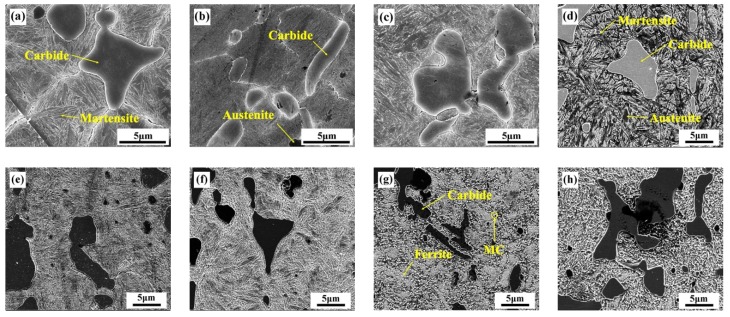
FESEM Micrograph of M42 HSS under Different Tempering Temperatures: (**a**) 200 °C, (**b**) 300 °C, (**c**) 400 °C, (**d**) 500 °C, (**e**) 550 °C, (**f**) 600 °C, (**g**) 650 °C, (**h**) 700 °C.

**Figure 2 materials-12-03714-f002:**
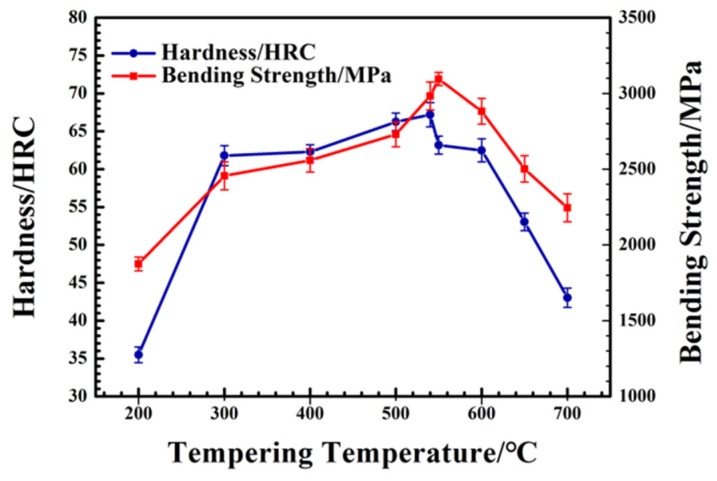
Hardness and Bending Strength of M42 HSS under Different Tempering Temperatures.

**Figure 3 materials-12-03714-f003:**
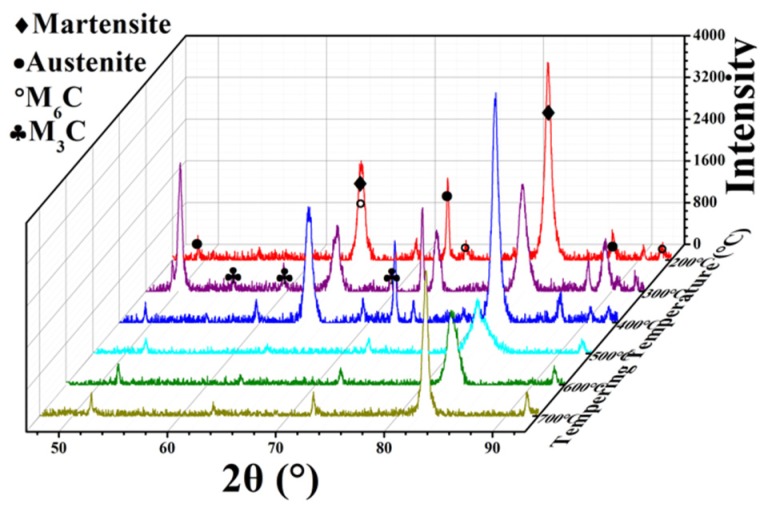
X-ray diffraction (XRD) patterns of M42 HSS after Different Tempering Treatments.

**Figure 4 materials-12-03714-f004:**
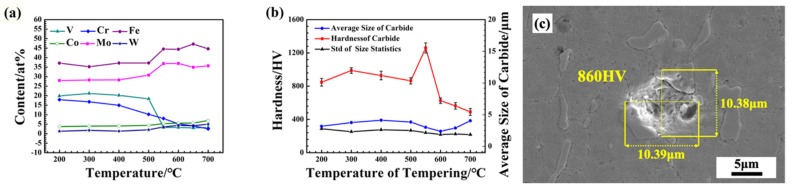
(**a**) Alloy Content of Carbide after Different Tempering Treatments. (**b**) Hardness and Average Size of Carbides after Different Tempering Treatments. (**c**) Microhardness micrograph of M42 after tempering at 500 °C.

**Figure 5 materials-12-03714-f005:**
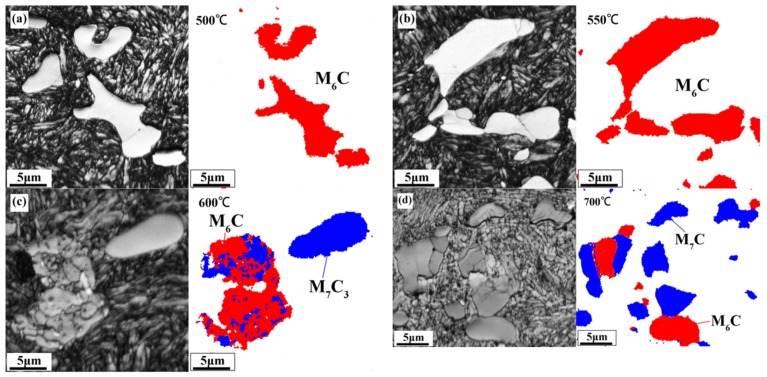
EBSD Band-Contrast Image and Phase Map of M42 HSS under different Tempering Temperatures: (**a**) 500 °C; (**b**) 550 °C; (**c**) 600 °C; (**d**) 700 °C.

**Figure 6 materials-12-03714-f006:**
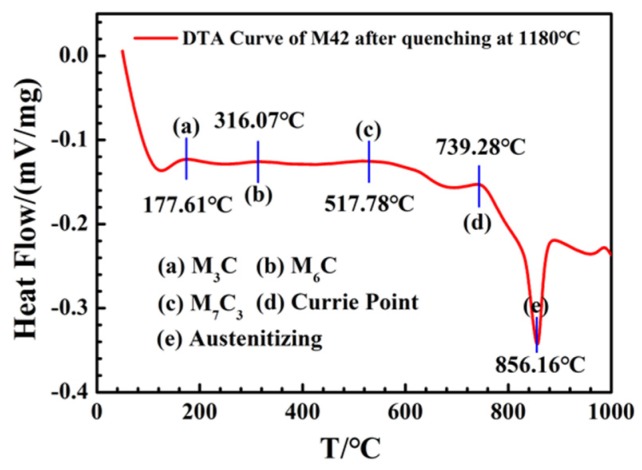
Differential Scanning Calorimetry (DSC) curve of M42 HSS after Quenching at 1180 °C.

**Figure 7 materials-12-03714-f007:**
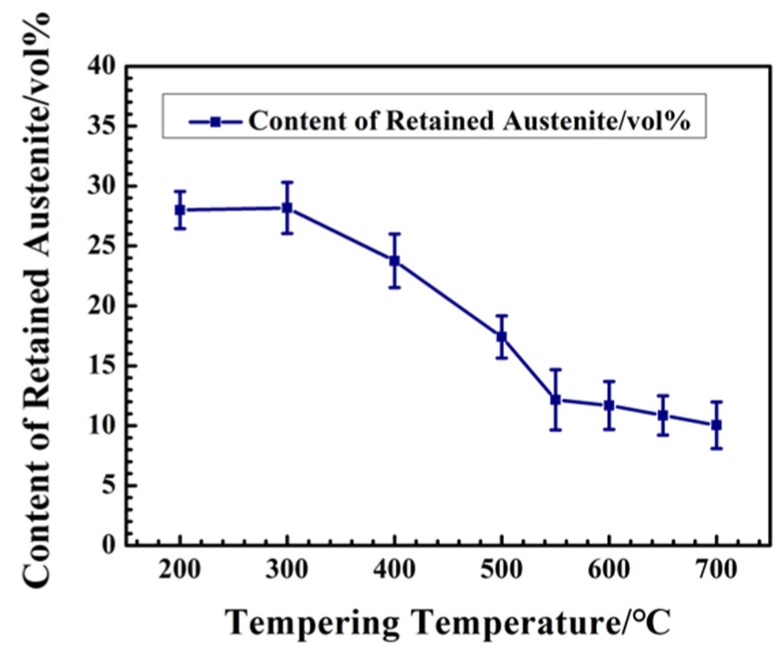
Content of Retained Austenite of M42 under different Tempering Temperatures.

**Figure 8 materials-12-03714-f008:**
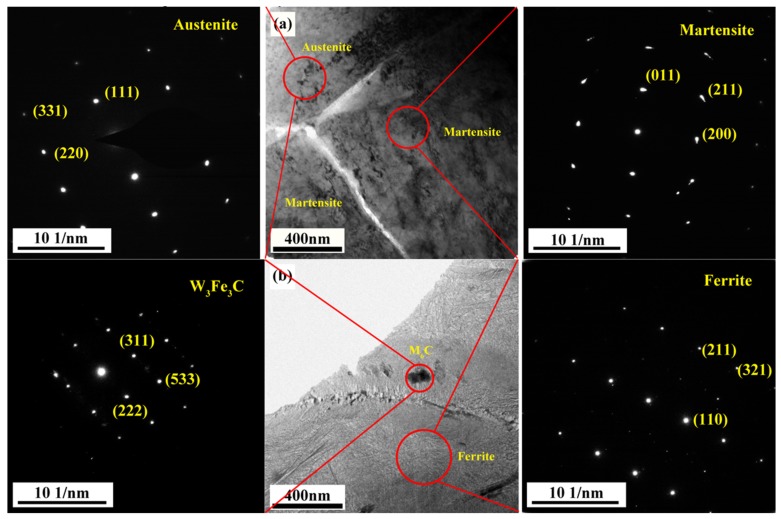
TEM and Selected Area Electron Diffraction Diagram of M42 HSS under different Tempering Temperatures: (**a**) 550 °C, (**b**) 700 °C.

**Table 1 materials-12-03714-t001:** Element contents of M42 High-Speed Steel (wt %).

C	Si	Mn	Mo	W	V	Co	Cr	Fe
1.10	0.25	0.36	9.50	1.50	1.15	8.00	3.75	Bal.

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
