# Peer review of "Effect of Tempering Conditions on Secondary Hardening of Carbides and Retained Austenite in Spray-Formed M42 High-Speed Steel"

_materials, 2019, doi:10.3390/ma12223714_

Round 1

Reviewer 1 Report

The submitted manuscript entitled ‘Effect of Tempering Conditions on Secondary Hardening Carbides and Retained Austenite of Spray Formed M42 HSS’ deals with the investigation of a specific high speed (HS) steel, with respect to the conditions of its tempering. The manuscript is interesting and sounds; however, during its review a list of issues and lacking information arose as listed below.

- Please provide an official e-mail address (instead of commercial) for all the Authors, but at least for the corresponding Authors.

- Please always let a space between the value and its unit, except in the case of ‘°C’ and ‘%’.

- Do not use abbreviations in the title and please solve every abbreviation at its first occurrence, even if they are well known and even if they are in the Abstract.

- Please mention the applied mechanical testing methods in the Abstract.

- Please provide the EU designation of the investigated steel in the form of HS-W-Mo-V-Co (all alloying elements in wt%).

- Please do not use ‘φ’ instead of ‘Ø’ to designate diameter.

- How was the chemical composition of table 1 measured?

- Please provide details about the oil quenching.

- If the cooling after the tempering was done in air, what is the aim of Ar atmospheres during the tempering?

- Page 3, line 97: V_C instead of the second V_A.

- Please identify the phases in fig 1. In the opinion of this Reviewer, the magnification is not sufficient to have a general impression about the microstructure.

- Please provide details about the circumstances (machine, setup, speed, etc.) of the bending tests of the investigated materials.

- Please do not start a section (3.3, 3.4) by a plot.

- Please provide details about the microhardness tests aimed to determine the hardness of the carbide. Please provide micrographs as well.

- Please add scatter to the measured (calculated) values of fig 7. The rest austenite amount of HSs can be decreased by overcooling (-70°C) after the quenching of the steel, did the Authors do any effort to decrease the rest austenite content by this simple method?

- The Authors list 19 references in sum at the end of their manuscript, that s simply considered insufficient regarding the properties of the HSs and their history in the professional literature. Please add more (at least additional 10) relevant references.

Author Response

Response to editor and reviewers’ comments

Manuscript title: Effect of Tempering Conditions on Secondary Hardening Carbides and Retained Austenite of Spray Formed M42 HSS.

Authors: Bowen Liu, Tian Qin, Wei Xu, Chengchang Jia, Qiuchi Wu, Mingying Chen, Zhe Liu

Manuscript ID: materials-642110

The authors would like to thank the editor and the reviewers for their valuable comments, which helped us to improve the quality of the manuscript and to clarify some key points. Our answers to the editor and reviewers’ comments are as follows. Changes in the revised manuscript are highlighted in yellow.

Reviewer 1

Comments: The submitted manuscript entitled ‘Effect of Tempering Conditions on Secondary Hardening Carbides and Retained Austenite of Spray Formed M42 HSS’ deals with the investigation of a specific high speed (HS) steel, with respect to the conditions of its tempering. The manuscript is interesting and sounds; however, during its review a list of issues and lacking information arose as listed below.

Response: We sincerely appreciate the thorough and attentive review. We have carefully revised the manuscript based on the comments.

Comment 1: Please provide an official e-mail address (instead of commercial) for all the Authors, but at least for the corresponding Authors.

Response: The e-mail address of the corresponding Author has been revised in the revised manuscript (line 10:*Correspondence). The official e-mail address of the other authors are as follow Table 1.

Table 1. Official e-mail address for all Authors.

Comment 2: Please always let a space between the value and its unit, except in the case of ‘°C’ and ‘%’.

Response: We have deleted the space between the value and °C and revised the manuscript.

Comment 3: Do not use abbreviations in the title and please solve every abbreviation at its first occurrence, even if they are well known and even if they are in the Abstract.

Response: We have revised all the abbreviations “HSS” to “high speed steel” in title and abstract, and solved all the abbreviation at its first occurrence in the revised manuscript.

Comment 4: Please mention the applied mechanical testing methods in the Abstract.

Response: We have added the applied mechanical testing methods in the Abstract. (line:15-16)

Comment 5: Please provide the EU designation of the investigated steel in the form of HS-W-Mo-V-Co (all alloying elements in wt%).

Response: M42 is America AISI designation of the steel investigated, and its EU designation is HS 2-9-1-8. We have revised the manuscript. (line:48)

Comment 6: Please do not use ‘φ’ instead of ‘Ø’ to designate diameter

Response: The ‘φ’ has been replaced by ‘Ø’ to designate diameter and revised in the manuscript (line:80).

Comment 7: How was the chemical composition of table 1 measured.

Response: The spray formed M42 high speed steel samples were supplied by Advanced Technology & Materials Co. China, and the chemical composition was also provided by the manufacturer. And the information has been added in the revised manuscript (line:75-76).

Comment 8: Please provide details about the oil quenching.

Response: Revised in manuscript (line:86-89). We have provided the detail quenching process as follow:

“The sample was oil quenched, and the detail process: embedded in alumina powders and in argon atmosphere to ensure uniform heating and avoid oxidation and decarburization, heated to 850℃ at 10 ℃/min keeping for 30 min, then heated to 1180℃ keeping for 5min, and finally oil quenched, cooling to room temperature”.

Comment 9: If the cooling after the tempering was done in air, what is the aim of Ar atmospheres during the tempering?

Response: Thanks for your valuable comment. There was a mistake in description for the tempering process. The tempering samples were cooling in Ar atmospheres until to room temperature. We have revised in manuscript (line:90-92).

Comment 10: Page 3, line 97: V_C instead of the second V_A

Response: Thanks for your valuable comment. The manuscript was revised (line:112).

Comment 11: Please identify the phases in fig 1. In the opinion of this Reviewer, the magnification is not sufficient to have a general impression about the microstructure.

Response: Thanks for your valuable comment. The manuscript was revised (Fig.1). According to your kind suggestion, we have defined some phases of Fig.1 to help us analyze the changes of the microstructure. But the repeated phases are not all defined. At the same time, we also revised the expression in 3.1, as shown in revised manuscript. This part is mainly used to observe the change of matrix with the increase of tempering temperature. Considering the change of martensite grain size and the transformation from martensite to ferrite, we choose a larger magnification to analyze the microstructure. (line:127-130, Fig.1)

Comment 12: Please provide details about the circumstances (machine, setup, speed, etc.) of the bending tests of the investigated materials.

Response: The bending strength test was carried out on the INSTRON universal material test machine at room temperature, with the sample size: 3 mm×3 mm×20 mm, and the deformation rate was 0.002 s-1. And the details about the circumstances was added in the revised manuscript (line:103-106).

Comment 13: Please do not start a section (3.3, 3.4) by a plot.

Response: Thanks for your valuable comment. The manuscript was revised (3.1, 3.3, and 3.4).

Comment 14: Please provide details about the microhardness tests aimed to determine the hardness of the carbide. Please provide micrographs as well.

Response: The microhardness of carbide was carried out on 401MVD-VD683 Wolpert Wilson Instruments, and the zone including carbide of M42 was tested. And the manuscript was revised (line:106-108).

Comment 15: Please add scatter to the measured (calculated) values of fig 7. The rest austenite amount of HSs can be decreased by overcooling (-70°C) after the quenching of the steel, did the Authors do any effort to decrease the rest austenite content by this simple method?

Response: Thanks for your valuable comment. The error bar of the measured (calculated) values of retained austenite has been added as shown in Fig.7. It can be seen from the other research that overcooling treatment is indeed an effective process to reduce the content of retained austenite. However, in industrial production, overcooling treatment will undoubtedly increase the cost of production. Based on the spray formed M42 high speed steel, the influence of tempering conditions on the secondary hardening of M42 carbide, and the influence of tempering temperature on the content of retained austenite were studied. The results show that the appropriate tempering process can effectively reduce the content of retained austenite, and the mechanism is explained from the perspective of microstructure transformation. Thus, we mainly studied the influence of tempering process on M42 high speed steel, but did not make further research on cryogenic treatment. We hope that in the future research, we can further combine the cryogenic treatment to prepare high speed steel products with more excellent properties.

Comment 16: The Authors list 19 references in sum at the end of their manuscript, that simply considered insufficient regarding the properties of the HSs and their history in the professional literature. Please add more (at least additional 10) relevant references.

Response: Thanks for your valuable comment. We have revised the “Introduction”, and added some new references including some latest ones.

Reviewer 2 Report

The reviewer suggests minor revisions of the present paper implying:The Authors should provide some more references from the last two years.In the paper there is a lack of information about procedure or standard for bend strength evaluation. Moreover, number of hardness test (points) should be provided. In Figure 2 we have HRC and Figure 4 - HV. In the “experiment” we have information about HRC. Thus, the Authors should provide info. about hardness tests measurements. Type of hardness tester machine for HRC and HV should be presented.The references for Va formula should be provided. In the paper information about practical application of these results should be presented.

Author Response

Response to editor and reviewers’ comments

Manuscript title: Effect of Tempering Conditions on Secondary Hardening Carbides and Retained Austenite of Spray Formed M42 HSS.

Authors: Bowen Liu, Tian Qin, Wei Xu, Chengchang Jia, Qiuchi Wu, Mingying Chen, Zhe Liu

Manuscript ID: materials-642110

The authors would like to thank the editor and the reviewers for their valuable comments, which helped us to improve the quality of the manuscript and to clarify some key points. Our answers to the editor and reviewers’ comments are as follows. Changes in the revised manuscript are highlighted in yellow.

Reviewer 2

Comments: The reviewer suggests minor revisions of the present paper implying: The Authors should provide some more references from the last two years. In the paper there is a lack of information about procedure or standard for bend strength evaluation. Moreover, number of hardness test (points) should be provided. In Figure 2 we have HRC and Figure 4 - HV. In the “experiment” we have information about HRC. Thus, the Authors should provide info. about hardness tests measurements. Type of hardness tester machine for HRC and HV should be presented. The references for Va formula should be provided. In the paper information about practical application of these results should be presented.

Response: We sincerely appreciate the thorough and attentive review. We have carefully revised the manuscript based on the comments.

(1) More references from the last two years have been added, thus we have revised the “Introduction”.

(2) The values for bend strength evaluation and hardness have been added. For the high speed steel samples used in this paper, the samples sold by the cooperative enterprise shall have a Rockwell hardness of more than 65 HRC and a bending strength of more than 3000 MPa. (line:317-319)

(3) 5 points for each samples were tested in hardness and microhardness testing. And the manuscript has been revised. (line:108)

(4) The details information about hardness tester machine for HRC and HV have been provided in “Experiment”. (line:103-107).

(5) The calculation formula of retained austenite volume fraction is derived from the national standard of China “GB 8362-87”. And we added the information in “Experiment” part (line:109-110).

Round 2

Reviewer 1 Report

Thank you for all the changes and corrections.